# Anchoring zero valence single atoms of nickel and iron on graphdiyne for hydrogen evolution

Yurui Xue[1], Bolong Huang [2], Yuanping Yi [1,3], Yuan Guo[1], Zicheng Zuo[1], Yongjun Li [1,3], Zhiyu Jia[1], Huibiao Liu[1,3] & Yuliang Li[1,3]

Electrocatalysis by atomic catalysts is a major focus of chemical and energy conversion effort. Although transition-metal-based bulk electrocatalysts for electrochemical application on energy conversion processes have been reported frequently, anchoring the stable transition-metal atoms (e.g. nickel and iron) still remains a practical challenge. Here we report a strategy for fabrication of ACs comprising only isolated nickel/iron atoms anchored on graphdiyne. Our findings identify the very narrow size distributions of both nickel (1.23 Å) and iron (1.02 Å), typical sizes of single-atom nickel and iron. The precision of this method motivates us to develop a general approach in the field of single-atom transition-metal catalysis. Such atomic catalysts have high catalytic activity and stability for hydrogen evolution reactions.

[1] Key Laboratory of Organic Solids, Institute of Chemistry, The Chinese Academy of Sciences, Beijing 100190, PR China. [2] Department of Applied Biology and Chemical Technology, The Hong Kong Polytechnic University—Hung Hom, Kowloon, Hong Kong SAR, China. [3] University of Chinese Academy of Sciences, Beijing 100049, PR China. Correspondence and requests for materials should be addressed to Y.L. (email: ylli@iccas.ac.cn)

The behavior of extrinsic atoms sitting on a surface of two-dimensional material can be metaphorically transformed into a floating boat on the immense sea. The local morphology of the surface as well as the binding between host and absorbates are seen as a key to stably locating these extrinsic atoms. The valence charge density overlaps between orbitals are indeed the essential media delivering the information between these two objects. Atomic catalysts (ACs) are expected to provide a critical approach to utilize every single metal atom and offer the most promising way to explore cost-effective catalysts[1–18]. The homogeneity of the active sites in ACs may fundamentally bridge the gap between heterogeneous and homogeneous catalysis, providing deep insight into the nature of the heterogeneous catalysis at an atomic level. Recent success in synthesizing ACs unambiguously demonstrated the fundamental and practical importance of ACs and has aroused great scientific interests in this new frontier of heterogeneous catalysis. But practical difficulties are serious obstacles for their scalable fabrication and real applications.

The prerequisite for practical application of ACs is successful anchoring single atoms on supports—a significant challenge due to the natural aggregation tendency of single atoms. Several strategies have emerged to cope with this challenge such as decreasing the loading amount of metal, tuning the metal-support interactions, and introducing vacancies or defects on/in supports. However, practical difficulties (e.g., rigorous synthetic conditions, process complexity and difficulty, specific instrument requirements, hard quality control, and low growth rate) have severely limited their development toward real applications. Moreover, reported ACs mainly focused on dispersing noble-metal atoms (Pt, Au, and Pd) on supports (metal oxides[2,11–13], metals[3], and graphene[8,9,16]). These are serious obstacles for scalable fabrication and real applications. Electrochemical deposition offers a convenient and efficient method to reduce metal ions into their elemental state. Unfortunately, up to now, no efforts have been made by electrochemical means for synthesizing ACs. Given these limitations, we became interested in developing new alternative supporting materials that are mechanically stable and can readily and effectively anchor isolated single atoms strongly enough to prevent aggregation under operating conditions.

Graphdiyne (GD)[19,20], a one-atom-thick two-dimensional carbon material with natural uniform pores and triple bond rich with strong reduction ability, has been applied in various research fields (such as catalysis[21,22], lithium-ion batteries[23,24], and solar cells[25,26]) due to its fascinating properties (e.g., excellent semiconducting properties, superior electrical conductivity, and mechanical stability). In GD, adjacent benzene rings ($sp^2$-hybridized carbon) linked to each other through butadiyne linkages (–C≡C–C≡C–, $sp$-hybridized carbon). In addition to $sp^2$ hybridization, the $sp$ hybridization of –C≡C– enables the arbitrary angle rotation of $\pi/\pi^\star$ perpendicular to the axis, making it possible to point toward metal atoms and, thereby, chelate a single metal atom[27]. Accordingly, GD may be an ideal candidate for direct use as the AC support without any pretreatment and provide a better opportunity to fabricate stable ACs. Transition metal [TM, such as nickel (Ni) and iron (Fe)] would discard the noble-metal usage and possess high utilization potential for industry[28]. We suspect that single-TM-atom-modified GD structure might be an excellent alternative to conventional single-atom catalysts.

Here we show that the GD can be direct use as the AC support without any pretreatment and provide an excellent opportunity to fabricate stable ACs (Ni/GD and Fe/GD) by a fast, scalable, controllable, and efficient electrochemical synthesis method that can be scaled readily. Our findings identify the very narrow size distribution of the Ni (1.23 Å) and Fe (1.02 Å) and these ACs

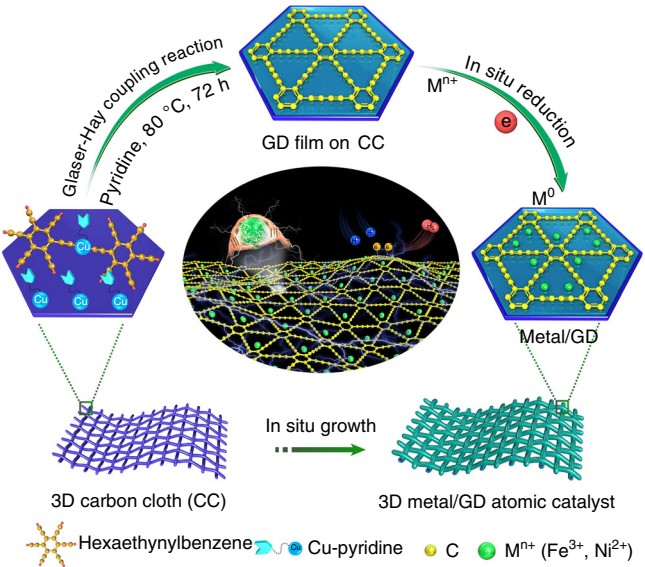

**Fig. 1** Protocols for the synthesis of Ni/GD and Fe/GD. A two-step strategy for anchoring-isolated Ni/Fe atoms on GD, including the in situ growth of GD layers on 3D carbon cloth (CC) surfaces via Glaser-Hay cross-coupling reaction, followed by the electrochemical reduction of metal ions ($Ni^{2+}$ and $Fe^{3+}$) into zerovalent metallic species [Ni(0) and Fe(0), respectively]

exhibit low onset overpotentials (close to 0 mV), small Tafel slopes, large turnover frequencies (TOFs), and high long-term stability among all reported nonprecious hydrogen evolution reaction (HER) single-atom catalysts and most of the state-of-the-art bulk catalysts—the result of strong chemical interactions and electronic coupling between the single TM atoms and GD that permit high degrees of charge transport between the catalytic active sites and the supports.

## Results

**Preparation and structural characterization.** The Ni/GD and Fe/GD ACs were prepared through a two-step strategy (Fig. 1), including the first preparation of three-dimensional (3D) GD foam (GDF) on carbon cloth (CC) surfaces via an acetylenic cross-coupling reaction using hexaethynylbenzene (HEB) as precursor[26], followed by the anchoring of Ni/Fe atoms by a facile electrochemical reduction method. It was found that the smooth of the CC was covered with a thin film of porous GD (Supplementary Fig. 1). We examined the qualities of the samples using X-ray photoelectron spectroscopy (XPS) and Raman spectroscopy. Compared with GDF (Supplementary Fig. 2, Supplementary Note 1), the C 1s XPS peaks of both Ni/GD and Fe/GD (Supplementary Fig. 3) all featured a satellite at 290.0 eV arising from the π–π* transition, which could be due to the restoration of the conjugated structure, indicative of interactions between Ni/Fe atoms and GD. The $sp/sp^2$ hybridization ratio remained equal to 2, suggesting that the anchoring of Ni/Fe atoms occurred without breaking any covalent bonds. From Raman spectra (Supplementary Fig. 4), the diffraction peaks of the diyne groups of Ni/GD and Fe/GD shifted slightly, consistent with the formation of chemical bonds after TM atom anchoring. The ratio of the D and G band intensities for both Ni/GD (0.87) and Fe/GD (0.85) are larger than that of GDF (0.77), suggesting more defects had formed, forming more active sites and finally improving the catalytic efficiency. Inductively coupled plasma mass spectrometry (ICP-MS) analysis indicated a Ni loading of 0.278 wt% on Ni/GD and Fe loading of 0.680 wt% on Fe/GD.

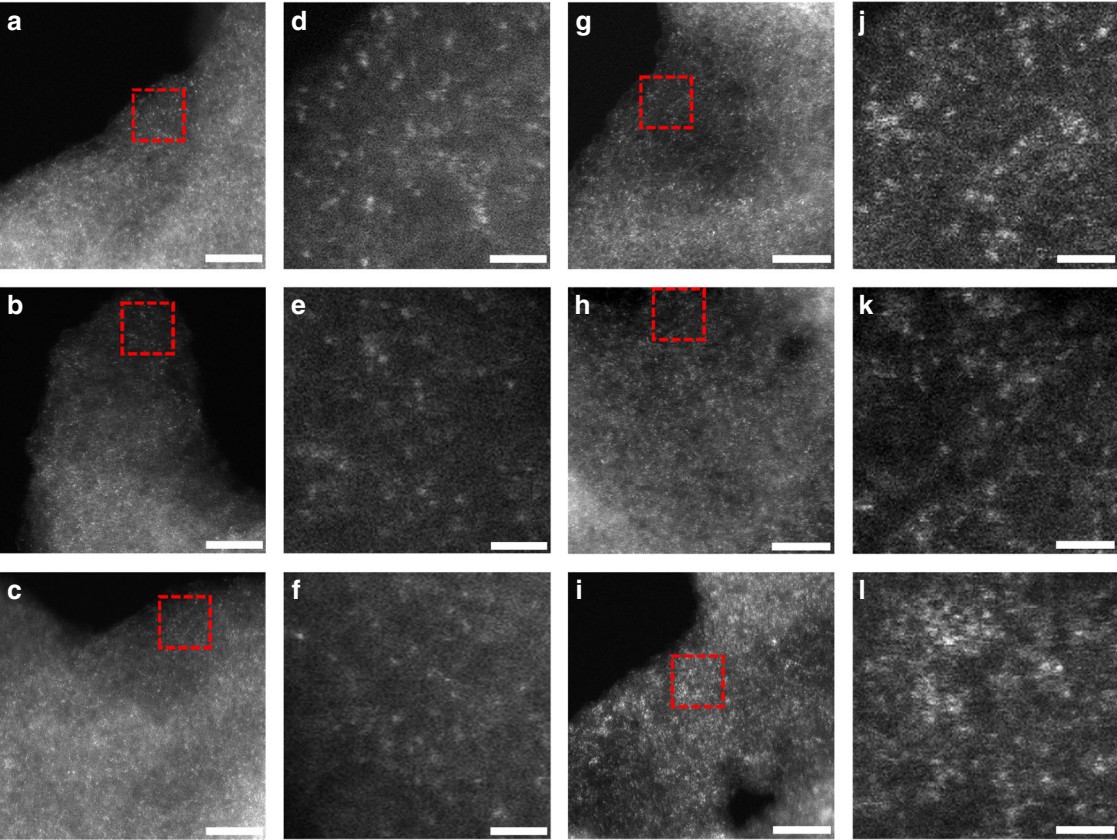

**Fig. 2** HAADF-STEM images of Ni/GD and Fe/GD. **a–c** Examination of various regions reveal that only isolated Ni atoms (white dots) are present and uniformly dispersed on the GD. Scale bars, 5 nm. **d-f** Corresponding enlargement of the marked regions in **a-c**. Scale bars, 1 nm. **g–i** Examination of various regions reveal that only isolated Fe atoms (white dots) are present and uniformly dispersed on the GD. Scale bars, 5 nm. **j-l** Corresponding enlargement of the marked regions in **g–i**. Scale bars, 1 nm

Sub-Ångström-resolution, aberration-corrected scanning transmission electron microscopy (STEM) was used for morphological characterization of Ni/GD and Fe/GD. Individual heavy atoms can be clearly evident in the atomic resolution high-angle annular dark-field (HAADF) images[2,8–12,29]. For Ni/GD, Fig. 2a–f clearly shows the uniform dispersion of isolated Ni atoms (white dots) on the surface of GD. Examination of various regions strongly confirms the only presence of single Ni atoms (Fig. 2a–f, Supplementary Figs. 5 and 6). For Fe/GD (Fig. 2g–l, Supplementary Fig. 7), HAADF images recorded at different regions clearly reveal that only isolated Fe atoms (white dots) are present and uniformly dispersed in Fe/GD. No formation of metal aggregates were observed in XPS (Supplementary Fig. 3), scanning electron microscope (SEM) images (Supplementary Fig. 8), transmission electron microscopy (TEM) images (Supplementary Fig. 8), and X-ray diffraction (XRD; Supplementary Fig. 9) analysis of the as-prepared catalysts. Additional HAADF-STEM images (Fig. 3a for Ni/GD and Fig. 3d for Fe/GD) and energy-dispersive X-ray spectroscopy analysis in a STEM revealed the uniform dispersion of Ni (Fig. 3b) and Fe (Fig. 3e), respectively. The histogram analysis shows very narrow size distributions of both Ni (1.23 ± 0.40 Å, inset of Fig. 3a) and Fe (1.02 ± 0.33 Å, inset of Fig. 3d), typical sizes of Ni and Fe atoms. These experiment results again demonstrate the reproducibility and universality of our strategy for synthesizing ACs.

To verify that the as-prepared ACs contained only atomically dispersed Ni/Fe atoms, X-ray absorption near-edge structure (XANES) and extended X-ray absorption fine structure (EXAFS) spectra were measured (Supplementary Fig. 10). For Ni/GD, there was only one notable peak at ~1.6 Å from the Ni ~ C contribution

and no peak in the region 2–3 Å from the Ni–Ni contribution (Fig. 3c), strongly confirming that Ni exists predominantly as isolated atoms. For Fe/GD, there was only one notable peak at ~1.5 Å from the Fe ~ C contribution for Fe/GD (Fig. 3f) and no Fe–Fe contribution (at ~2.2 Å) was observed, confirming the only presence of singly dispersed Fe atoms in catalyst (Supplementary Table 1). The derivative XANES indicates Fe/GD (or Ni/GD) consists of an Fe(0) [or Ni(0)] center (Supplementary Note 2). To gain a deeper insight into the states of Ni (Fe) atoms in Ni/GD (Fe/GD), XANES and EXAFS spectra were measured after hydrogen reduction (5% $H_2$/He) at different temperatures. Each temperature step is maintained for 30 min. We did not find any change in the XANES and EXAFS spectra before and after hydrogen reduction for Ni/GD and Fe/GD (Supplementary Figs. 11 and 12). The results showed that the Ni and Fe atoms in Ni/GD and Fe/GD still were metallic state, respectively.

**Theoretical studies**. The geometry of absorption site has further been studied (Fig. 4a). First, for the Ni/Fe adatoms absorbing on a single GD layer, the binding energies (Supplementary Table 2) show that the site S1 for Ni/Fe (−3.72/−1.22 eV, respectively) were more energetic favorable than the site S2 (−1.39/−0.37 eV). A strong chemisorption character has been found through bond length analysis and the presence of strong charge transfer from Ni/Fe to GD (Fig. 4b). Further for the intercalation of TM between GD layers[30,31], the potential energy surface scanning along two possible pathways (route A and B) is performed (Supplementary Fig. 13). The Ni/Fe are inclined to absorb along route A and the most favorable adsorption site is at A1 (−1.98 and −1.66 eV for Fe and Ni, respectively).

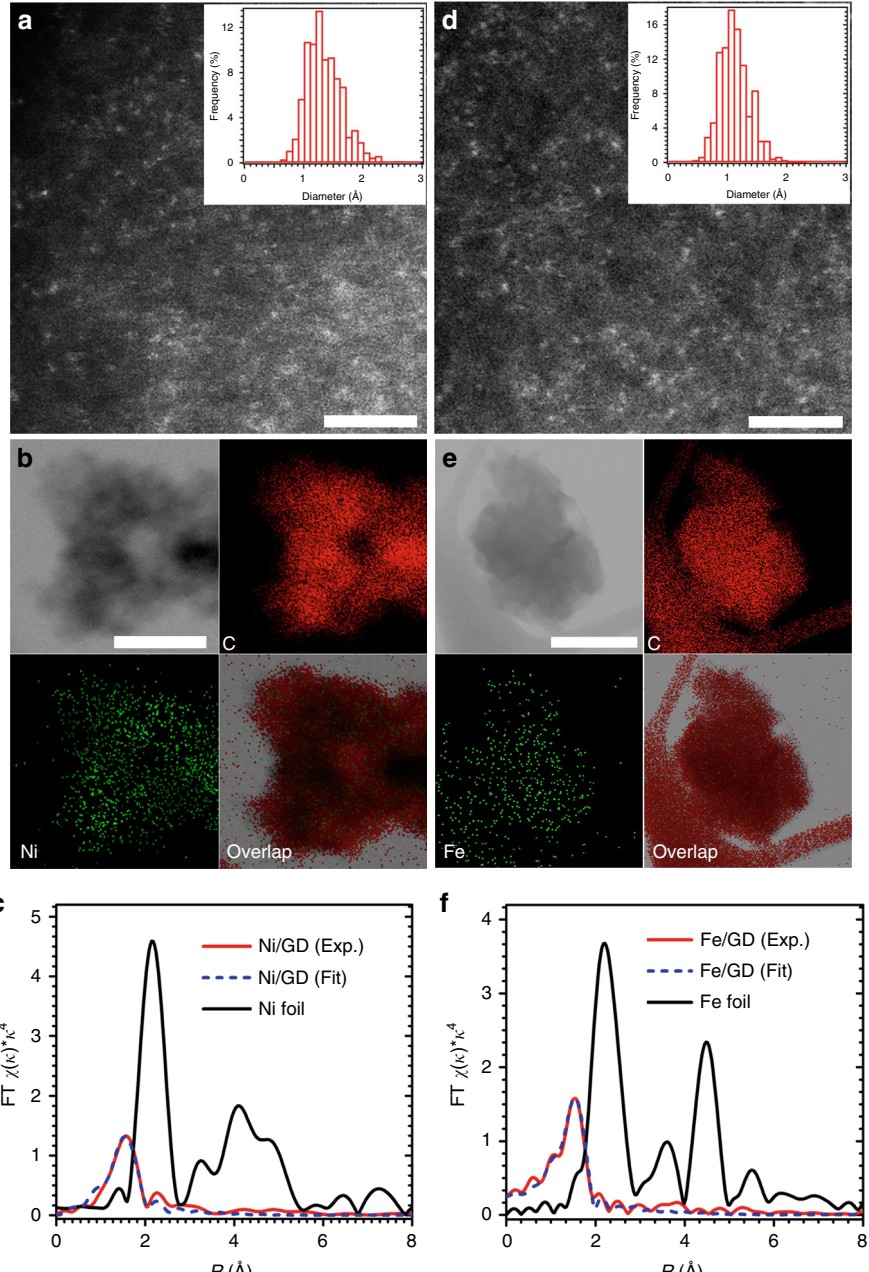

**Fig. 3** XAS studies and size distribution. **a** Additional HAADF-STEM image of Ni/GD [inset: size distribution of Ni atoms counted from HAADF-STEM images (>560 Ni atoms considered, the most probable value is 1.23 ± 0.40 Å)]. Scale bar, 2 nm. **b** STEM elemental mapping of Ni/GD. Scale bar, 200 nm. **c** Ex situ EXAFS spectra of Ni/GD and Ni foil at the Ni K-edge. **d** Additional HAADF-STEM image of Fe/GD [inset: size distribution of Fe atoms counted from HAADF-STEM images (>1070 Fe atoms considered, the most probable value is 1.02 ± 0.33 Å)]. Scale bar, 2 nm. **e** STEM elemental mapping of Fe/GD. Scale bar, 200 nm. **f** Ex situ EXAFS spectra of Fe/GD and Fe foil at the Fe K-edge. All atoms counted in the HAADF-STEM images were well separated from their neighbors

To elucidate the hidden information on subtle metal atoms–C bonding, we utilized previously developed method to ab initially reflect the energy of the targeted on-site orbital especially with electronic occupations under different cases of chemical bonding[32–35]. Let's take the Ni-3d as an example. We reflected the open-shell effect (non-crossover) in cubic-NiO (Fig. 4c), and see the closed-shell effect (crossover) but with minor 3d–3d overlaps in fcc-Ni (Fig. 4d). Differently, the Ni-3d orbital on the GD also exhibits a closed-shell effect but larger orbital potential implying stronger orbital overlaps between Ni and neighboring C sites (Fig. 4e). We further extend the computational efforts on the Ni-3d orbital energy variation with related to the change of Ni–C

bond length (Fig. 4f). The different location sites of Ni within GD, S1, and S2 sites are tested respectively with related to the variations of Ni–C inter-distances. The Ni–C distance dependent Ni-3d orbital potential crossover points are illustrated. We find that the S1 and S2 site orbital energy variations define the boarder lines of the most probable closed-shell orbital energy of Ni acting as $Ni^0$ state under the thermodynamic equilibrium adsorption state. Within this region, our EXAFS measured data point, S1 site, and S2 site are all staying in the same horizontal line pointing to the same orbital potential energy. This means the stabilized Ni has nearly the same orbital energy regardless the short-range (nearest neighboring) or even medium-range (second nearest

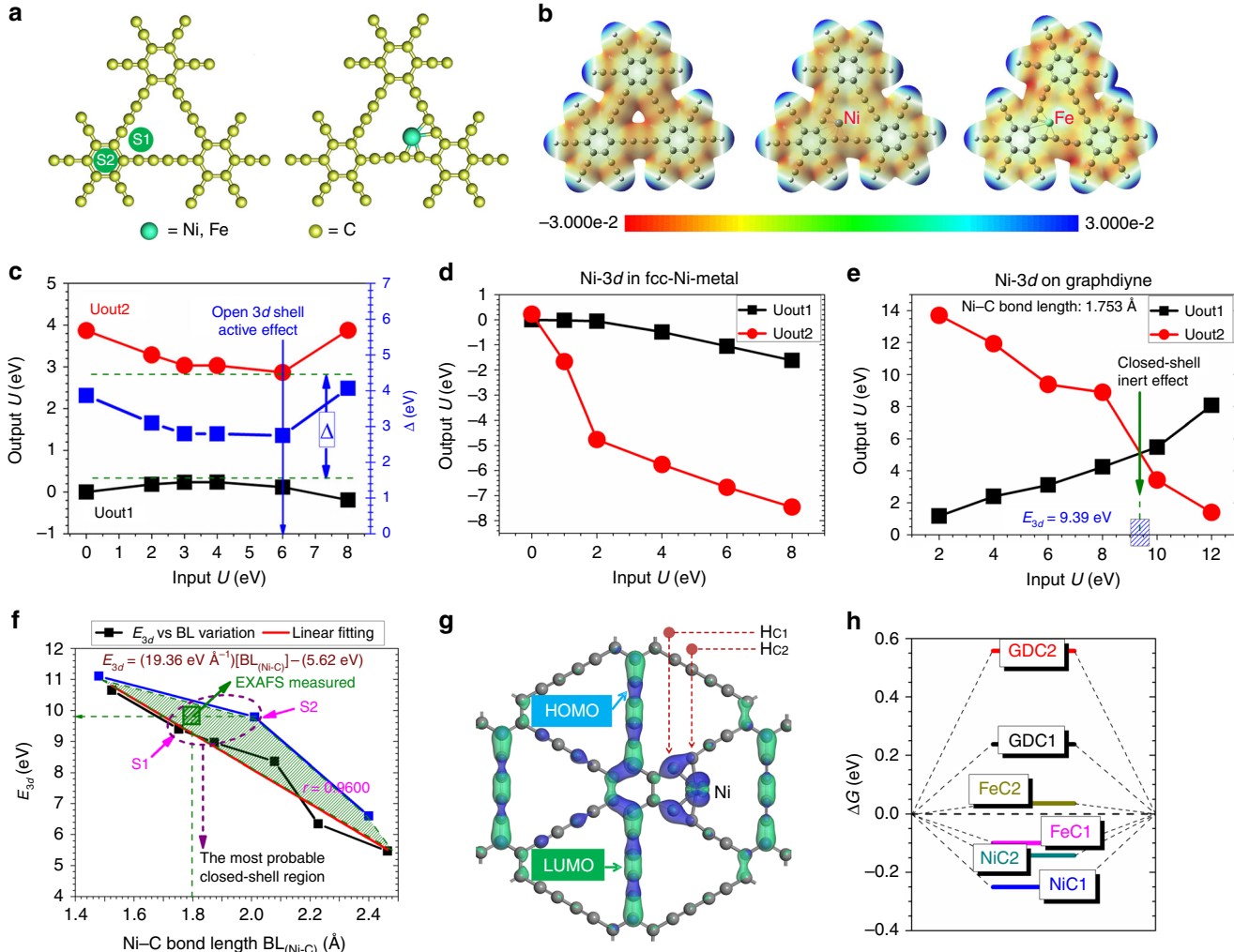

**Fig. 4** Theoretical studies. **a** Adsorption of single metal atoms on GD (left: possible adsorption sites; right: optimized configuration). **b** Electrostatic potential maps of pristine GD, Ni/GD, and Fe/GD, respectively. The 3d orbital energies for the targeted Ni site in **c** NiO, **d** Ni-on-GD, and **e** Ni-fcc are self-consistently determined via open- and closed-shell charge overlapping, respectively. **f** The variation of orbital energy variation with related to the newly formed Ni–C. The green shaded area denotes the Ni–C inter-distance-dependent Ni-closed-shell regions with related to Ni–C inter-distance. The purple dashed circle shows the most probable closed-shell orbital regions for Ni to be located on GD system in thermodynamic equilibrium state. The green square tells our experimental EXAFS measured data. **g** Real-space HOMO and LUMO contour plots on Ni-on-GD. The $H_{C1}$ and $H_{C2}$ denote the active H adsorption site on different C sites labeled with C1 and C2, respectively. **h** The chemisorption energy of H for HER performance with related to the free energy profile ($\Delta G$). The Ni/FeC1 and Ni/FeC2 mean the H adsorption on the C1 and C2 sites within Ni/Fe-on-GD system, respectively. The GDC1 and GDC2 denote the H adsorption on pristine GD system

neighboring) Ni–C environments within Ni-on-GD system. The S1 site shows more stable with 1.15 eV less than the S2 site for Ni, while their 3d orbital energies are nearly the same as ~9.6 eV. Therefore, the Ni–C bonding in this system could be only understood as orbital charge overlaps rather than the realistic covalent/ionic bond defined conventionally. We have illustrated the abnormal strong (sp)-d overlapping induces inert closed-shell effect for TM on GD and lead to an extrinsic charge compensation. Thus, the neutral state for Ni on GD is not as same as the intrinsic neutral state. Then the oxidation state of pristine $Ni^{2+}$ turns to $Ni^0$ compensated by the evident Ni(3d)-C (sp) overlaps. Accordingly, it is analogous to consistently update the Ni site with $3d^{8+\delta}$ ($\delta \sim 2$), where the $\delta$ equivalently compensates the charge depletion given by the $4s^2$ ionization. But where does this additional orbital charge $\delta$ come from? Note, the GD surface in fact exists with strong electronegative sp-hybridized orbitals localizing near the C-ring. The overlapping between C-(2s2p) and Ni-3d orbital induces the charge

occupation to be mutually compensated. The electronic structural calculation shows that the (sp)-d orbital overlapping strongly compensate the active 3d orbital of Ni site, and passivate them into an inert shell. The valence compensation in fact undergoes charge transfers via inter-orbitals instead of inter-sites. The Ni–C bond length is optimized as 1.753 Å close to the measured 1.8 Å (Fig. 3c). The Ni-3d orbital energies are self-consistently obtained as 9.79, in agreement with measured 9.89 eV by EXAFS. From Fig. 4f, the Ni-3d orbital energies are self-consistently obtained as 9.79 eV for S1 and 9.39 for S2, in agreement with measured 9.89 eV. We are now clearly seeing the HOMO and LUMO charge densitiy distributions (Fig. 4g) induced by the stabilized $Ni^0$ and able to determine the active sites for fast ($H^+$ +e) charge exchanges that are responsible for the efficient HER reaction. We then move onto the HER performance via illustrating the hydrogen chemisorption energy diagram in terms of the changes in Gibbs free energy (Fig. 4h)[21]. In general, the energetic interval between the chemisorption level and the thermoneutral

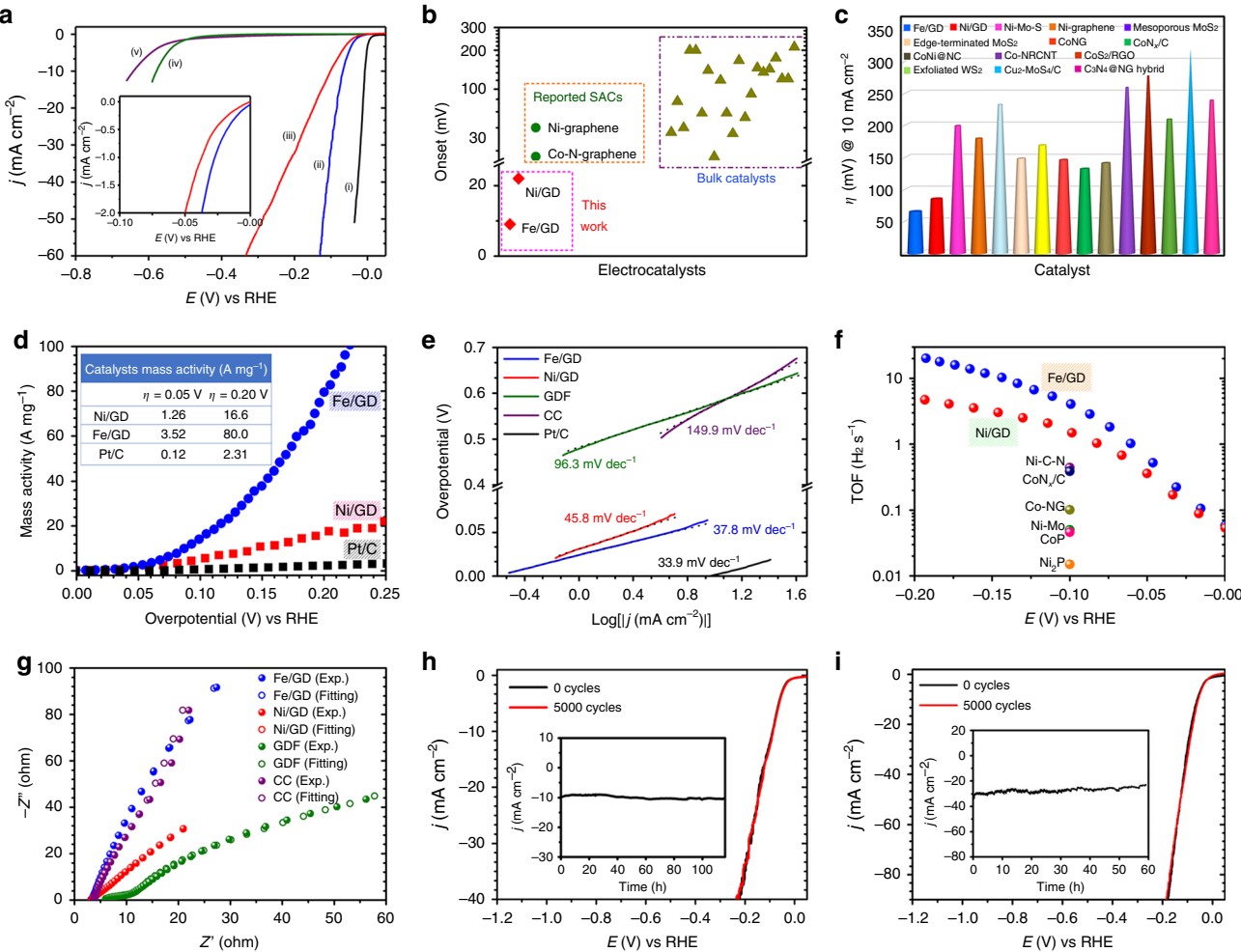

**Fig. 5** HER activities and stabilities of Ni/GD and Fe/GD. **a** Polarization curves of (i) Pt/C, (ii) Fe/GD, (iii) Ni/GD, (iv) GDF, and (v) CC (inset: enlarged view of the LSV curves for Fe/GD and Ni/GD near the onset region). **b** Onset values and **c** overpotentials at 10 mA cm$^{-2}$ of Ni/GD and Fe/GD (red square) along with other nonprecious single-atom HER catalysts (green circle) and several bulk catalysts (olivine triangle). **d** Mass activities of Ni/GD, Fe/GD, and Pt/C (inset: mass activities obtained at overpotentials of 0.05 and 0.20 V). **e** Tafel plots of the presented data in **a**. **f** TOF values of Fe/GD (blue dot) and Ni/GD (red dot) together with several state-of-the-art HER electrocatalysts. **g** Nyquist plots of experimental data (dots) and fitting results (circles) for Fe/GD, Ni/GD, GDF, and CC. Stability tests of **h** Ni/GD and **i** Fe/GD (insets: respective time-dependent current density curves)

line ($\Delta G = 0$ eV) determines the abilities of absorption/desorption of H$^+$/H$^0$. The deeper the stronger absorption but rather weaker in desorption, vice versa. Different electronically active C sites in Ni/Fe-on-GD system show that the HER performance of Ni/Fe-on-GD is superior to the pristine GD as it promotes both the initial adsorption (rate-determining step) and final desorption. The HER performance of Fe could be better than the case of Ni as the Ni-on-GD might have over-binding effect to reduce the efficiency of H$^0$ desorption. Thus, lower Ni or higher Fe coverage might reach the optimal HER performance.

Based on the characterization data mentioned above, it is evident that both Ni/GD and Fe/GD consist of only isolated Ni and Fe atoms, respectively. The strong interaction between the single TM atoms and the GD is, on one hand, beneficial to stabilize the single metal atoms on the GD support. In addition, the strong charge transfer from Ni/Fe to GD (Supplementary Table 2) was also considered to be beneficial for the electrical conductivity and catalytic activity.

**HER catalytic activity of ACs.** We chose the HER, which is a critical process for renewable-energy storage and conversion device[36–42], as a probe reaction to investigate the catalytic

performance of ACs. After excluding the effects of scanning rate on HER performance (Supplementary Fig. 14), we evaluated the HER catalytic activity of the Fe/GD and Ni/GD ACs in H$_2$-saturated 0.5 M H$_2$SO$_4$ using a standard three-electrode system (the reference electrodes were calibrated daily with respect to reversible hydrogen electrode (RHE), Supplementary Fig. 15). GDF, CC, and Pt/C (20 wt%) were used for comparison. In Fig. 5a, the rapid rise of current with the increasing of cathodic potential showed that Fe/GD and Ni/GD can act as high-performance electrodes for HER. Fe/GD exhibits the best HER activity with the smallest onset overpotential of 9 mV, which is very close to Pt/C (1.0 mV, Supplementary Fig. 16). Fe/GD shows an overpotential of 66 mV at 10 mA cm$^{-2}$, which is smaller than that of Ni/GD (88 mV), GDF (578 mV), and CC (642 mV) (Fig. 5a–c). The values are superior to other earth-abundant HER electrocatalysts [such as Ni$_2$P (~110 mA cm$^{-2}$)[43], MoS$_2$ (170 mA cm$^{-2}$)[44], and exfoliated WS$_2$ (210 mA cm$^{-2}$)[45]], and even compare favorably with noble-metal-based electrocatalysts such as Pt nanowires/single-layered Ni(OH)$_2$[46]. Mass activity is one of the important evaluation criteria applied to characterize the catalytic performances in practical applications[47]. Remarkably, normalized to respective loading, both Fe/GD and Ni/GD possess significantly better mass activity toward HER than commercial Pt/C

(Fig. 5d), for example, at overpotential of 0.2 V, the mass activities of Fe/GD (80.0 A mg$_{metal}^{-1}$) and Ni/GD (16.6 A mg$_{metal}^{-1}$) are 34.6 and 7.19 times greater than that of Pt/C (2.31 A mg$_{metal}^{-1}$), respectively. The XANES and EXAFS spectra (Supplementary Fig. 17) for Fe/GD and Ni/GD showed there are no changes before and after HER measurements. The result indicated that the structures of Fe/GD and Ni/GD ACs were very stable.

The Tafel slopes provide further insights into the HER mechanism. As shown in Fig. 5e, Fe/GD and Ni/GD showed very small Tafel slopes of 37.8 and 45.8 mV dec$^{-1}$, respectively, which are only little larger than that of Pt/C (33.9 mV dec$^{-1}$). The Tafel slopes suggested the Volmer–Heyrovsky mechanism was operative during the HER process. These values are superior to all reported nonprecious single-atom catalysts and most of the state-of-the-art HER catalysts such as CoP (50 mV dec$^{-1}$)[48] (Supplementary Tables 3 and 4). Compared with the pristine GDF (onset overpotential: 430 mV; Tafel slope: 96.3 mV dec$^{-1}$), the Ni/GD and Fe/GD exhibited high HER activity. This is due to the associated interactions between single Ni/Fe atoms and GD. The geometrical exchange current density ($j_{0,geometrical}$) is another inherent measure of activity for HER. The magnitude of $j_0$ is measure of rate of reaction. Higher the $j_{0,geometrical}$ the faster is the reaction rate. The $j_{0,geometrical}$ for Ni/GD (0.25 mA cm$^{-2}$) and Fe/GD (0.29 mA cm$^{-2}$) are much larger than reported ACs [e.g., Ni-doped graphene (5.3 × 10$^{-2}$ mA cm$^{-2}$)[8]] and most of the bulk HER catalysts such as CoP (0.14 mA cm$^{-2}$)[48] (Supplementary Table 4), revealing the exceptional intrinsic HER activities of Fe/GD and Ni/GD ACs. The effects of deposition time and concentration on HER performance are presented in Supplementary Figs. 18 and 19. Fe and Ni nanoparticles with large size decorated GD were synthesized and characterized, which showed much lower HER activities compared to Fe/GD and Ni/GD ACs (Supplementary Figs. 20–23) further demonstrated the superior activity of the as-synthesized ACs.

In catalysis, TOF is the best figure of merit to compare the intrinsic activities of catalysts with different surface areas or loadings[41]. Assuming that all single metal atoms in the catalyst are active and accessible to the electrolyte, the upper limit number of the active sites for Ni/GD and Fe/GD were estimated to be 2.56 × 10$^{16}$ and 2.38 × 10$^{16}$ sites per cm$^2$, respectively, which are more than that of Pt(111) (1.5 × 10$^{15}$ sites per cm$^2$)[36]. The TOF of Ni/GD and Fe/GD at 100 mV were calculated to be 1.59 and 4.15 s$^{-1}$ (Fig. 5f), respectively, which are much higher than other reported state-of-the-art electrocatalysts, such as CoP (0.046 s$^{-1}$)[47] and Ni$_2$P (0.015 s$^{-1}$)[43] (Supplementary Table 5).

The electrochemical impedance spectroscopy (EIS) data for catalysts are fitted to an equivalent circuit model (Supplementary Fig. 24). Compared with GDF and CC (Fig. 5g and Supplementary Table 6), the smaller solution resistance ($R_s$) and charge-transfer resistance ($R_{ct}$) of Fe/GD and Ni/GD indicate the improved charge-transfer behavior over the catalyst interface[8]. Besides, the lower hydrogen adsorption resistance for Fe/GD (1.19 Ω) and Ni/GD (3.63 Ω) reveals the more favorable absorption of the hydrogen intermediates, which is substantially beneficial for the overall electrocatalytic performance.

The electrochemical active surface area (ECSA) of each AC was estimated by determining the double-layer capacitance ($C_{dl}$) (see Methods section for details)[42,49,50]. The $C_{dl}$ for Fe/GD, Ni/GD, GDF, and CC are 0.63, 0.29, 0.19, and 0.08 mF, respectively. Both Fe/GD and Ni/GD show significantly increased $C_{dl}$ as compared with the CC support, indicating that the $C_{dl}$ is actually contributed from the active single atoms. For our system, specific capacitance of $C_s = 0.035$ mF cm$^{-2}$ was used to calculated the ECSA[42]. The ECSA for Fe/GD, Ni/GD, GDF, and CC are 18, 8.3, 5.4, and 2.3 cm$^2$, respectively. The specific current densities ($j_s$)

for our ACs were further calculated. At $\eta = -100$ mV, Fe/GD and Ni/GD show $j_s$ of 0.21 and 0.12 mA cm$^{-2}$, respectively.

Stability is a critically important property of all catalysts. The stability of Ni/GD and Fe/GD was estimated by running long-term potential cycling tests and potential-constant electrolysis. The polarization curves of Ni/GD (Fig. 5h) and Fe/GD (Fig. 5i) are still unchanged in current density after 5000 cycling tests, which are much more superior on stability to commercial Pt/C (Supplementary Fig. 25). Besides, Ni/GD and Fe/GD showed only negligible losses in $j$ even after 116 and 60 h of constant electrolysis, respectively, which could be ascribed to the hindrance from the adsorbed hydrogen bubbles. As evidenced by XPS and HAADF-STEM characterizations (Supplementary Figs. 26 and 27), no changes in the composition occurred and both Ni and Fe atoms were presented as isolated atoms without any aggregations for Ni/GD and Fe/GD during the long-term cycling operation. Maximum Faradaic efficiencies around 98% for Fe/GD and Ni/GD were achieved (Supplementary Fig. 28). Moreover, to check the leaching to the solution during the electrochemical measurements, ICP-MS analysis for direct trace metal determinations in electrolyte before and after electrochemical tests were conducted, detecting almost no decrease in metal species, which further reflects the stability of the ACs. Based on these findings, we ascribe the outstanding stability of such ACs to chemically inert, mechanically stable property of GD and the strong chemical bonding of single Ni/Fe atoms to the surrounding carbon atoms of GD.

## Discussion

We report the facile and precise anchoring of the single-atom Ni (0) and Fe(0) on GD by electrochemical reduction of Ni$^{2+}$ and Fe$^{3+}$ on GD. We have shown that GD can be used as a powerful support for the anchoring of single-atom Ni and Fe catalysts. Our findings identify very size distribution of the single-atom Ni and Fe mainly around 1.23 and 1.02 Å, typical sizes of single Ni and Fe atoms, respectively. These GD-supported ACs exhibit extremely high catalytic activity and long-term stability for HER under acidic conditions. Their performances are superior to that all reported ACs and the most state-of-the-art bulk non-precious catalysts. Such outstanding HER performances originated from strong chemical interactions and electronic coupling between the single-atom Ni/Fe and GD that permit high degrees of charge transport between the catalytic active sites and the supports. Considering the significantly improved simplicity, large-scalability, low cost, precision, and accuracy of this synthesis strategy, our approach developed here appears to be a simpler and more designable way of fabricating highly efficient ACs, while also possibly paves a path for their practical commercialization.

## Methods

**Materials**. All reagents were commercially available as analytical grade. All water used was purified with a Millipore system (typically 18.2 MΩ cm resistivity). The CC was pretreated by sonication sequentially in concentrated nitric acid, deionized water, acetone, ethanol, and deionized water before use. The freshly pretreated CC and copper foils were used immediately for the preparation of GDF. HEB precursor was synthesized and purified according to ref. [20].

**Synthesis of GDF**. The 3D CC and Cu foils, closely stacked (the Cu foil on the top and the CC on the bottom), were put into the same reactor containing pyridine. Cu foils in pyridine (an alkaline solution) can easily release Cu ions into solution and subsequently interact with pyridine molecules, forming Cu-pyridine complex that can easily diffuse onto the template surface and catalyze the coupling of alkynyls. HEB solution (10 mM in pyridine) was then added to the reaction flask in which CC was fixed closely to Cu foils. The mixture was kept at 80 °C for 3 days under Ar atmosphere. The products were then thoroughly washed[20]. A flexible 3D GDF having a porous surface was obtained (GDF). The color changes from light gray to black, indicating the complete coverage of GD film on CC.

**Synthesis of Ni/GD.** The freshly prepared GDF was immersed in $H_2SO_4$ aqueous solution (500 mM) containing $NiSO_4$ (5 mM) to allow the adsorption of Ni species. Electrochemical deposition of Ni was conducted under galvanostatic conditions at $10\ mA\ cm^{-2}$ for 150 s using CHI 760E electrochemical Workstation at room temperature. The obtained Ni/GD was then thoroughly cleaned and immediately used for electrochemical measurements.

**Synthesis of Fe/GD.** Fe/GD was synthesized similar to that of Ni/GD but with minor modifications. The freshly prepared GDF was immersed in $H_2SO_4$ aqueous solution (500 mM) containing $FeCl_3$ (10 mM) to allow the adsorption of Fe species. Fe electrodeposition was conducted at $10\ mA\ cm^{-2}$ for 250 s at room temperature. After a thorough washing, the obtained Fe/GD was immediately used for electrochemical measurements.

**Characterization.** SEM and TEM/high-resolution TEM images were recorded on a field-emission SEM (Hitachi S-4800) and a 200 kV JEOL-2100F microscope, respectively. Energy-dispersive X-ray spectroscopy (EDX) analysis was performed with an energy-dispersive X-ray detector in Hitachi S-4800 SEM. The powder XRD experiments were carried out with a PANalytical high-resolution XRD system (model EMPYREAN) using Cu Kα radiation ($\lambda = 0.15406$ nm). XPS was performed on Kratos Axis Ultra DLD with monochromated Al Kα radiation ($hv = 1486.6$ eV). Raman spectra were collected with a Renishaw-2000 Raman spectrometer.

**HAADF-STEM characterizations.** HAADF-STEM images were obtained on the aberration-corrected cubed FEI Titan Cubed Themis G2 300 or JEM-ARM200F (JEOL, Tokyo, Japan) TEM/STEM operated at 200 kV with cold filed-emission gun and double hexapole Cs correctors (CEOS GmbH, Heidelberg, Germany). The attainable spatial resolution defined by the probe-forming objective lens is better than 80 pm. HAADF-STEM imaging was also.

**X-ray absorption spectroscopy measurements and data processing.** X-ray absorption spectroscopy (XAS) measurements were performed at the 1W1B beamline of the Beijing Synchrotron Radiation Facility. The XAS raw data were background-subtracted, normalized, and Fourier-transformed by standard procedures with the ATHENA program[51,52]. Least-squares curve fitting analysis of the EXAFS $\chi(k)$ data, including multiple shell contributions was carried out using the ARTEMIS program with the theoretical scattering amplitudes, phase shifts, and the photoelectron mean free path for all paths calculated by the ab initio code FEFF 6.0. The data were fitted in R-space.

**Calculation setup.** The electronic properties have been described with recently developed ab initio orbital corrections[32–35]. The core of the method reflects a generalized searching path toward the optimal parameters valid to exhibits the different chemical bonding information within various materials systems[32,33]. Here in the work, total energy calculation can be successfully achieved with consideration of the targeted orbital projected under any given case of bonding.

This method has been imbedded within simple DFT + U[8] coding complied in the CASTEP source package[53]. The orbital $U$ parameter for Ni-3d is ab initially searched[54] based on the geometry-optimized structure through Broyden–Fletcher–Goldfarb–Shannon algorithm with Perdew–Burke–Ernzerhof (PBE) functional. The planewave basis set cutoff energy is determined at 750 eV with $k$-point mesh of $4 \times 4 \times 2$ after convergence test. To prevent spin-charge vibration during the electronic minimization process, ensemble density functional theory (DFT)[55] is accounted instead of block Davidson-based density mixing technique under the same reciprocal space[56]. With Hellmann–Feynman theorem, the necessary tolerence for total energy calculation convegerging is set to $5.0 \times 10^{-7}$ eV per atom to guarantee the lattice ionic forces are approaching to the level of 0.001 eV Å$^{-1}$.

Scalar relativistic averaged pseudopotentials for Ni and C are considered with KB-projector and partial core correction accounted[57–59]. The normal local electronic exchange-correlation-induced overlapping effect in PBE between valence and core has been alleviated with our chosen method[60,61]. To reduce demanding computational cost, the reduced (3d, 4s, and 4p) and (2s and 2p) KB-projectors are represented for configuring the RRKJ-framework pseudopotential of Ni and C, respectively, in the generation[62] for matching the planewave basis cutoff energy.

The geometric structures for the adsorption of metal atoms on GD were optimized by DFT at the B3LYP/6-31G($d$,$p$) level. The binding energies were calculated with the basis set superposition error corrected further.

**Electrochemical measurements.** Electrochemical measurements were performed a three-electrode system (CHI. 760E, Shanghai CH. Instruments, China). The as-prepared samples, saturated calomel electrode (SCE), and graphite plate were used as the working electrode, reference electrode, and counter electrode, respectively. Polarization curves were collected in 0.5 M $H_2SO_4$ at 0.5 mV s$^{-1}$. The electrolyte has been degassed by $H_2$ before starting the experiment. Cyclic voltammogram measurements were conducted within $-0.6$ to 0 V at 100 mV s$^{-1}$. All final potentials converted to RHE values. The EIS measurements were investigated in

the frequency from 100 kHz to 0.1 Hz. In our experiments, SCE was calibrated daily with respect to RHE (Supplementary Fig. 15).

**Calculation of TOF and active sites.** The TOF value per nickel site was calculated according to the following equation:

$$\text{TOF} = \frac{\#\text{Total hydrogen turnovers per geometric area}}{\#\text{Acitve sites per geometric area}} \qquad (1)$$

The total number of hydrogen turnovers was calculated from the current density:

$$\#\text{Total hydrogen turnovers} = \left(j\ \frac{\text{mA}}{\text{cm}^2}\right)\left(\frac{1\text{C/s}}{1000\ \text{mA}}\right)\left(\frac{1\text{mol}\ e^-}{96485.3\text{C}}\right)\left(\frac{1\text{mol}\ H_2}{2\text{mol}\ e^-}\right)\left(\frac{6.022 \times 10^{23}\ \text{molecules}\ H_2}{1\ \text{mol}\ H_2}\right)$$
$$= 3.12 \times 10^{15}\ \frac{H_2/s}{cm^2}\ \text{per}\ \frac{mA}{cm^2} \qquad (2)$$

The upper limit number of the actives was calculated based on the hypothesis that all single metal atoms (Ni and Fe) on GD form the active centers, and are accessible to the electrolyte. Their contents (Ni: 0.278 wt%; Fe: 0.68%) were calculated from the ICP-MS data.

For example,

$$\#\text{Active sites(Ni)} = \left(\frac{\text{Ni wt}\% \times \text{catalyst loading per geometric area (g/cm}^2)}{\text{Ni Mw (g/mol)}}\right)\left(\frac{6.022 \times 10^{23}}{1\ \text{mol}\ \text{Ni}}\right)$$
$$= \left(\frac{0.278\% \times 0.896 \times 10^{-3}\ (\text{g/cm}^2)}{58.69\ (\text{g/mol})}\right)\left(\frac{6.022 \times 10^{23}}{1\ \text{mol}\ \text{Ni}}\right) = 2.56 \times 10^{16}\ \text{sites}\ \text{cm}^{-2} \qquad (3)$$

Finally, the current density from the LSV polarization curve can be converted into TOF values according to:

$$\text{TOF}_{\text{Ni/GD}} = \left(\frac{3.12 \times 10^{15}}{2.56 \times 10^{16}} \times j\right) = 0.1219 \times j \qquad (4)$$

Similarly, for Fe/GD,

$$\#\text{Active sites (Fe)} = \left(\frac{\text{Fe wt}\% \times \text{catalyst loading per geometric area (g/cm}^2)}{\text{Fe Mw (g/mol)}}\right)\left(\frac{6.022 \times 10^{23}}{1\ \text{mol}\ \text{Ni}}\right)$$
$$= \left(\frac{0.680\% \times 0.325 \times 10^{-3}(\text{g/cm}^2)}{55.85\ (\text{g/mol})}\right)\left(\frac{6.022 \times 10^{23}}{1\ \text{mol}\ \text{Ni}}\right) = 2.38 \times 10^{16}\ \text{sites}\ \text{cm}^{-2} \qquad (5)$$
$$\text{Therefore, TOF}_{\text{Fe/GD}} = \left(\frac{3.12 \times 10^{15}}{2.38 \times 10^{16}} \times j\right) = 0.131 \times j$$

**Calculation of the mass activity.** The mass activity ($j_{\text{mass}}$) can be obtained according to the equation:

$$j_{\text{mass}} = \frac{j_{\text{geometrical}}}{M_{\text{loading}}} \qquad (6)$$

where $j_{\text{geometrical}}$ is the geometric activity of the catalysts, and $M_{\text{loading}}$ is the catalyst loading per geometric surface area. For Fe/GD and Ni/GD, the $M_{\text{loading}}$ are 0.00221 and 0.00249 mg cm$^{-2}$, respectively.

**Calculation of ECSA and $C_{\text{dl}}$ values.** The ECSA was estimated from the electrochemical $C_{\text{dl}}$ according to the equation:

$$\text{ECSA} = \frac{C_{\text{dl}}}{C_{\text{s}}} \qquad (7)$$

Here the $C_{\text{s}}$ is 0.035 mF cm$^{-2}$ based on reported values[42].

The double-layer capacitances used were measured using EIS. The electrochemical system is approximated by an equivalent circuit model, including $R_{\text{s}}$, a constant-phase element of the double-layer (CPE$_{\text{dl}}$), $R_{\text{ct}}$, and the constant-phase element of the adsorbed layer (CPE$_1$). The impedance of the CPE is given by equation

$$Z_{\text{CPE}} = Y^{-1}(i\omega)^{-n} \qquad (8)$$

where $Y$ is a proportional factor, $i = (-1)^{1/2}$, $\omega$ is the frequency of the sinusoidal applied potential, and $n$ is a factor satisfying the condition $1 \geq n \geq 0$.

The $C_{\text{dl}}$ was calculated according to the following equation

$$C_{\text{dl}} = (Y_{\text{dl}}R_{\text{ct}})^{1/n}/R_{\text{ct}} \qquad (9)$$

**Calculation of specific current densities ($j_s$).** $j_s$ can be calculated by dividing geometrical current density ($j_g$) at a given overpotential by the determined

roughness factor ($R_f$), which was calculated by dividing ECSA by geometric area of the electrode:

$$j_s = \frac{j_g}{R_f} \tag{10}$$

**Data availability**. The data that support the findings of this study are available from the corresponding author upon reasonable request.

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

## Acknowledgements

This work was supported by the National Nature Science Foundation of China (21790050 and 21790051), the National Key Research and Development Project of China (2016YFA0200104), and the Key Program of the Chinese Academy of Sciences (QYZDY-SSW-SLH015). We thank the XAFS station (beamline 1W1B) of the Beijing Synchrotron Radiation Facility. We also thank the Center for Electron Microscopy (Tianjin University of Technology) for high angle annular dark field (HAADF) imaging.

## Author contributions

Yuliang Li designed and supervised the research and analysis, reviewed and edited this manuscript. Y.X. performed the catalyst synthesis and characterizations, electrochemical experiments, collected and analyzed the data, and wrote the original draft of manuscript. B.H., Y.Y., and Y.G. performed the theoretical calculations. Z.Z., Yongjun Li, Z.J., and H. L. gave technical support and helpful advice.

## Additional information

**Competing interests:** The authors declare no competing interests.

