## [Peer Review File · Nature Communications]

Reviewers' comments:

Reviewer #1 (Remarks to the Author):

In their revised manuscript the authors really tried to address most of the requested points and therefore I can recommend it for publication in Nature Communication. They added several experiments and theoretical calculation to further support their claims and to elucidate the operation mechanism during the hydrogen evolution reaction. The reported material performances are very promising (mainly the stability in acid). I still think that the single atom doping advantages will be more pronounced for rare-metals if they can show it in order to increase the impact of their work. However, I think that this work can be published in its current form and in a future work the authors or other (upon reading this nice work) may consider doing so.

Reviewer #2 (Remarks to the Author):

Anchoring single-atom Ni(0) and Fe(0) on graphdiyne for hydrogen evolution

This manuscript reported the single-atom Ni(0) and Fe(0) dispersed on graphdiyne for hydrogen evolution reaction. An electrochemical deposition method was applied to introduce the Ni(0) and Fe(0) atom into graphdiyne structure. X-ray adsorption fine structure analysis and aberration-corrected high angle annular dark field scanning transmission electron microscopy are further performed and indicate that the chemical structure of Fe and Ni is predominantly composed of isolated metal atoms. From the electrochemical performance view, the HER overpotential of Fe/GD catalyst is relatively low among non-precious metal catalysts. As graphdiyne is a less explored material in electrocatalysis area, both its preparation and HER performance are interesting for a broad society of researchers. However, some concerns should be investigated and solved for the benefit of the readers. Considering the reasons above, I recommend the acceptance of the current manuscript in Nature Communication after major revisions.

1. More characterization should be performed on graphdiyne. Both low and high magnification TEM images of graphdiyne or Fe(Ni)/GD are required in Supplementary Figure 1. In Supplementary Figure 2d and Figure 4, the signal of Raman peaks at 1953 and 2189 cm^{-1} are accompanied by high intensity of background noise, which should be improved. It will be helpful to determine the average thickness of graphdiyne layer grown on the electrode. AFM or SEM characterization could be included if possible.

2. The Fe(0) and Ni(0) contained complex is generally air-sensitive or water-sensitive. It is possible that the Fe and Ni species are still present as Fe^{2+} or Ni^{2+} with graphdiyne coordination (like ferrocene). The authors' claim of "zerovalent metallic species" is questionable and not well-supported. Although some theoretical DFT calculations (which may not be reliable) have been made to explain this claim, more experimental measurements are required and should be performed. The pre-edge derivative of XANES spectra of Fe(Ni)/GD could provide more information about the metal's covalence state while comparing with Fe or Ni compound references (ACS Nano, 2017, 11, 6930-6941). The signal/noise ratio of XPS characterization on Fe and Ni needs to be improved if possible (Figure S3). If Fe or Ni species are not as Fe0 or Ni0 in Fe(Ni)/GD catalyst, the conclusion and title of this work must be corrected.

3. The quality of electrochemical measurement in Figure 5 is poor. In Figure 5a, the HER current density plot of Ni/GD is too noisy to be accepted. The sharp rise of cathodic current at $j > 50 \text{ mA cm}^{-2}$

in Pt/C result is strange and needs to be replotted. Enlarged view of HER currents near the onset region should be included in Figure 5 for clarity. In Figure 5d, if the mass activity is calculated by the mass of loading metal, the unit should mention it (such as A mgmetal^{-1}), and the details of calculation should be given. Moreover, the calibration curve of SCE reference electrode with RHE should be provided in the supporting information to validate the accuracy of electrochemical measurement.

4. On Page 14 line 244, the authors claimed that Fe/GD exhibits the smallest overpotential of only 9 mV. However, it is hard to see that in Figure 5. Even in Figure 5d, the onset potential is more likely at 50 mV. The authors should explain more about how this value of overpotential is obtained. Otherwise, the authors should correct this value and revise the phrase "close to 0 mV" in Page 4 line 76. The XANES characterization of Fe(Ni)/GD before and after HER measurements should also be performed in order to determine whether the atomic structure of Fe or Ni is changed during the HER measurement.

The detailed point-by-point responses to each comment are listed as follows:

Reviewer #1 (Remarks to the Author):

In their revised manuscript the authors really tried to address most of the requested points and therefore I can recommend it for publication in Nature Communication. They added several experiments and theoretical calculation to further support their claims and to elucidate the operation mechanism during the hydrogen evolution reaction. The reported material performances are very promising (mainly the stability in acid). I still think that the single atom doping advantages will be more pronounce for rare-metals if they can show it in order to increase the impact of their work. However, I think that this work can be published in its current form and in a future work the authors or other (upon reading this nice work) may consider doing so.

Response: Thank you very much for your very thoughtful review of our work. We really appreciate all of your comments for our manuscript.

Reviewer #2 (Remarks to the Author):

This manuscript reported the single-atom Ni(0) and Fe(0) dispersed on graphdiyne for hydrogen evolution reaction. An electrochemical deposition method was applied to introduce the Ni(0) and Fe(0) atom into graphdiyne structure. X-ray adsorption fine structure analysis and aberration-corrected high angle annular dark field scanning transmission electron microscopy are further performed and indicate that the chemical structure of Fe and Ni is predominantly composed of isolated metal atoms. From the electrochemical performance view, the HER overpotential of Fe/GD catalyst is relatively low among non-precious metal catalysts. As graphdiyne is a less explored material in electrocatalysis area, both its

preparation and HER performance are interesting for a broad society of researchers. However, some concerns should be investigated and solved for the benefit of the readers. Considering the reasons above, I recommend the acceptance of the current manuscript in Nature Communication after major revisions.

1. More characterization should be performed on graphdiyne. Both low and high magnification TEM images of graphdiyne or Fe(Ni)/GD are required in Supplementary Figure 1. In Supplementary Figure 2d and Figure 4, the signal of Raman peaks at 1953 and 2189 cm^{-1} are accompanied by high intensity of background noise, which should be improved. It will be helpful to determine the average thickness of graphdiyne layer grown on the electrode. AFM or SEM characterization could be included if possible.

Response: Thanks for your comments. Low-magnification TEM image of graphdiyne has been added in Supplementary Figure 1. Low-magnification TEM images of Ni/GD and Fe/GD have been provided in Supplementary Figure 7. The Raman spectra illustrated in Supplementary Figure 2d and Figure 4 have been smoothed to remove background noise. The average thickness of graphdiyne layer grown on the electrode has been determined using SEM characterization (typical SEM image is shown in Supplementary Figure 1f).

These have been added in the revised manuscript as Supplementary Figures as below.

Supplementary Figure 1: Morphology characterizations. (a) Low- and (b) high-magnification SEM images of CC. Low- and high-magnification (inset) SEM images of GDF (c). (d) Low- and (e) high-resolution TEM (HRTEM) images of GDF. (f) SEM image of GDF showing the layer thickness (~ 22 nm) measurements.

Supplementary Figure 2: XPS and Raman characterization of CC and GDF. High resolution XPS spectra of C 1s for pure CC (**a**) and GDF (**b**); Raman spectra of pure CC (**c**) and GDF (**d**).

Supplementary Figure 4: Raman characterization of Ni/GD and Fe/GD. Raman spectra of

Ni/GD (a) and Fe/GD (b). From Raman spectra, the diffraction peaks of the diyne groups of Ni/GD and Fe/GD shifted slightly, consistent with the formation of chemical bonds after TM atoms anchoring. The ratio of the D and G band intensities for both Ni/GD (0.87) and Fe/GD (0.85) are larger than that of GDF (0.77), suggesting more defects had formed, favoring the accessibility of more active sites and potentially increasing the catalytic efficiency.

Supplementary Figure 7: Morphology characterizations. Low- and high-magnification (inset) SEM images of Ni/GD (a) and Fe/GD (d), respectively. (b) Low- and (c) high-resolution TEM (HRTEM) images of Ni/GD. (e) Low- and (f) high-resolution TEM (HRTEM) images of Fe/GD.

2. The Fe(0) and Ni(0) contained complex is generally air-sensitive or water-sensitive. It is possible that the Fe and Ni species are still present as Fe^{2+} or Ni^{2+} with graphdiyne

coordination (like ferrocene). The authors' claim of "zerovalent metallic species" is questionable and not well-supported. Although some theoretical DFT calculations (which may not be reliable) have been made to explain this claim, more experimental measurements are required and should be performed. The pre-edge derivative of XANES spectra of Fe(Ni)/GD could provide more information about the metal's covalence state while comparing with Fe or Ni compound references (ACS Nano, 2017, 11, 6930-6941). The signal/noise ratio of XPS characterization on Fe and Ni needs to be improved if possible (Figure S3). If Fe or Ni species are not as Fe⁰ or Ni⁰ in Fe(Ni)/GD catalyst, the conclusion and title of this work must be corrected.

Response: The reviewer has put forward a good question for us to further understand our research work and make our work more perfect.

As discussed in the manuscript (please see the manuscript for details), the most favorable adsorption site for single metal atoms (e.g., Fe and Ni) on monolayer graphdiyne (GD) is all in the alkyne ring (**Fig. R1a**). Further for intercalation of single metal atoms between GD layers, the most favorable adsorption site for the Ni/Fe atoms is A1 (**Fig. R1b and R1c**), we can find the Ni/Fe atoms lies between two GD layers. The case is more like that of metallocenes (e.g., ferrocene and nickelocene), a type of organometallic compound consisting of a central metal atom between two cyclopentadienyl rings (*please see refs.: R1-R3*). Previous studies have provided the best understanding of the interaction between metal center and two organic unites. The metal can be regarded as in a zero oxidation state (*please see refs.: R4, R5*). This means that ferrocene consists of an Fe(0) center and two cyclopentadienyl ligands.

It is a very important evidence from previous literature concerning the metallocenes from which we can find the main peaks of ferrocene and nickelocene in two metallorganic compounds are mainly appeared around 7130 eV and 8344 eV, respectively (**Fig. R2**, *please see ref.: R6*). In our experimental XANES spectra (**Fig. R3**), the main peaks of Fe/GD and Ni/GD are located at 7125 eV and 8340 eV, respectively. It is obviously that both Fe/GD and

Ni/GD show the shifts of the adsorption edge to smaller energies compared to that of ferrocene and nickelocene, respectively. There is no doubt that the zero-valence atoms, compared to the zero-valence bulk materials and clusters (single atoms), the interaction may be more complex from that of their own. And confinement of electrons leads to a discrete energy level distributions and a distinctive HOMO-LUMO gap. These lead to quantum size and small size effects, resulting in the more change of XANES spectrum.

According to the above-discussed, it can be considered that Fe/GD (or Ni/GD) consists of an Fe(0) [or Ni(0)] center.

Figure R1. Diagrams of single metal atom-adsorbed GD. (a) The most favorable adsorption site for single metal atoms on monolayer GD. (b) Top view and (c) side view of single metal atoms adsorbed bilayer GD.

Figure R2. Comparison between the experimental XANES spectra (upper traces) and “computer generated” XANES spectra (lower traces) for ferrocene (a) and nickelocene (b). Symbols, A, B, C and D denote specific features detected in the experimental XANES spectra. (a) Ferrocene (lower) traces: (...) the primitive, polarization-averaged $X\alpha$ calculation, (---) the same $X\alpha$ calculation after convolution with a Gaussian-like energy broadening function (2.5 eV full width at half maximum), (—) calculation carried out with a complex Hedin-Lundqvist potential but without further convolution with the energy broadening function including the core-hole lifetime and the finite experimental resolution. (The arrow also indicates the energy location of the antibonding orbital $5e_{1g}$, as found in our calculation.) (b) Nickelocene (lower traces): (...) the primitive, polarization-averaged $X\alpha$ calculation, (---) the same $X\alpha$ calculation after convolution with a Gaussian-like energy broadening function (2.5 eV full width at half maximum). Reprinted with permission from ref R6. Copyright 1988 Elsevier.

Figure R3. (a) The normalized Fe K-edge XANES spectra and first derivative curves (the inset) of different samples and references. (b) The normalized Ni K-edge XANES spectra and first derivative curves (the inset) of different samples and references.

The following aspects should be helpful for understanding the formation and structural features of our catalysts.

(1) Graphdiyne is completely different from graphene on stabilizing metals. In graphdiyne (GD)

structure, each benzene ring is connected to six benzene rings through butadiyne linkages ($-C\equiv C-C\equiv C-$), rich triple-bond possess strong reduction ability to metal, which could be very beneficial to the reduction of metal ions and stable the zero-valent state of the metal.

The pore structure of these graphdiyne provides an ideal space to associate stable zero-valent metal. We must realize that graphdiyne is completely different from graphene on structure and chemical properties. It can stabilize zero-valent metals from its unique chemical properties. These are the inability of graphene and other carbon materials to do it.

I sincerely hope that the reviewer can understand this new idea. If I have the opportunity, I would like to have an in-depth discussion with the reviewer. As an innovative concept, the reviewer will understand it, thank him very much.

(2) DFT calculation is very important for understanding the zero valence state of the metal.

It is completely consistent with our experimental process and characterization. DFT calculation is universal and the most accurate calculation method at present. The DFT calculations all play a key role in designing novel catalysts. Examples can be found in the following literatures:

- Seh Z. W. et al. Combining theory and experiment in electrocatalysis: Insights into materials design. *Science* **355**, 146, eaad4998 (2017).
- Nørskov, J. K. et al. Towards the computational design of solid catalysts. *Nat. Chem.* **1**, 37-46 (2009).
- Greeley, J. et al. Computational high-throughput screening of electrocatalytic materials for hydrogen evolution. *Nat. Mater.* **5**, 909-913 (2006);
- Liu, P. et al. Photochemical route for synthesizing atomically dispersed palladium catalysts. *Science* **352**, 797-800 (2016).
- Deng, D. et al. Catalysis with two-dimensional materials and their heterostructures. *Nat. Nanotech.* **11**, 218-230 (2016).
- Zhang, B. et al. Homogeneously dispersed multimetal oxygen-evolving catalysts. *Science* **15**, 333-337 (2016).
- Calle-Vallejo, F. et al. Finding optimal surface sites on heterogeneous catalysts by counting nearest neighbors. *Science* **350**, 185-189 (2015).
- Greeley, J. et al. Alloys of platinum and early transition metals as oxygen reduction electrocatalysts. *Nat. Chem.* **1**, 552-556 (2009).

- (3) All atomic catalysts were prepared through a facile in situ electrochemical reduction procedure under a strong acidic condition (0.5 M H₂SO₄). In this process, metal ions could gain electrons and then be reduced to metal atoms with a zero valence. These Ni(0) and Fe(0) then anchored on the surface of graphdiyne.
- (4) To eliminate any possible contaminate (such as air or water), samples were immediately used for electrochemical measurements in 0.5 M H₂SO₄ after the preparation.
- (5) As discussed in our manuscript (Page 9 line 148-154), to gain a deeper insight into the states of Ni (Fe) atoms in Ni/GD (Fe/GD), X-ray absorption near-edge structure (XANES) and extended x-ray absorption fine structure (EXAFS) spectra of Ni/GD (Fe/GD) samples were in situ measured under hydrogen (5% H₂/He) atmosphere at different temperatures. Each temperature step is maintained for 30 minutes. Under this condition, metal species

at high valence state can be reduced to metal, therefore leading to obvious variation in the XANES and EXAFS spectra. Actually, there is no difference in XANES and EXAFS spectra were observed before and after hydrogen reduction for both Ni/GD and Fe/GD. These observations reflect the metallic state of the Ni and Fe atoms in Ni/GD and Fe/GD, respectively.

The signal noise of XPS characterization on Fe and Ni in Supplementary Figure 3 have been improved.

Supplementary Figure 3: XPS characterization of Ni/GD and Fe/GD. High resolution XPS spectra of C 1s (a), Ni 2p (b), and XPS survey spectra (c) of Ni/GD; High resolution XPS spectra of C 1s (d), Fe 2p (e), and XPS survey spectra (f) of Fe/GD.

Supplementary Figure 11: XAS studies of Ni/GD catalysts at different temperature under 5% H_2/He . **a**, *Ex situ* EXAFS spectra and **(b)** the normalized XANES spectra at the Ni K edge of Ni/GD at the Ni K-edge obtained at different reduction conditions. Ni foil was measured for comparison.

Supplementary Figure 12: XAS studies of Fe/GD catalysts at different temperature under 5% H_2/He . **a**, *Ex situ* EXAFS spectra and **(b)** the normalized XANES spectra at the Fe K edge of Fe/GD at the Fe K-edge obtained at different reduction conditions. Ni foil was measured for comparison.

References:

- R1. Kealy, T. J. & Pauson, P. L. A new type of organo-iron compound. *Nature* **168**, 1039-1040 (1951).
- R2. Wilkinson, G., Rosenblum, M., Whiting, M. C. & Woodward, R. B. The structure of iron bis-cyclopentadienyl. *J. Am. Chem. Soc.* **74**, 2125-2126 (1952).

- R3. Dunitz, J. D. & Orgel, L. E. Bis-cyclopentadienyl – a molecular sandwich. *Nature* **171**, 121-122 (1953).
- R4. Bell, C. F. Syntheses and physical studies of inorganic compounds. Pergamon: New York, 1972, Chapter 20.
- R5. Strohfeldt, K. A. Essentials of inorganic chemistry: for students of pharmacy, pharmaceutical sciences and medicinal chemistry. John Wiley & Sons, Ltd. 2015, Chapter 8.
- R6. Ruiz-Lopez, M. F., Loos, M., Goulon, J., Benfatto, M. & Natoli, C. R. Reinvestigation of the EXAFS and xanes spectra of ferrocene and nickelocene in the framework of the multiple scattering theory. *Chem. Phys.*, **121**, 419-437 (1988).

3. The quality of electrochemical measurement in Figure 5 is poor. In Figure 5a, the HER current density plot of Ni/GD is too noisy to be accepted. The sharp rise of cathodic current at $j > 50 \text{ mA cm}^{-2}$ in Pt/C result is strange and needs to be replotted. Enlarged view of HER currents near the onset region should be included in Figure 5 for clarity. In Figure 5d, if the mass activity is calculated by the mass of loading metal, the unit should mention it (such as $\text{A mg}_{\text{metal}}^{-1}$), and the details of calculation should be given. Moreover, the calibration curve of SCE reference electrode with RHE should be provided in the supporting information to validate the accuracy of electrochemical measurement.

Response: Thanks for your comments.

(1) The HER current density plot of Ni/GD has been rationally smoothed (Figure 5a in revised manuscript).

(2) New LSV curve for Pt/C (20 wt.%) was plotted in Figure 5a in revised manuscript.

(3) Enlarged view of HER currents near the onset region has been provided in Figure 5a in revised manuscript.

(4) The unit of the mass activity has been changed to $\text{A mg}_{\text{metal}}^{-1}$. The mass activity (j_{mass})

can be obtained according to the equation:

$$j_{mass} = \frac{j_{geometrical}}{M_{loading}}$$

where $j_{geometrical}$ is the geometric activity of the catalysts, and $M_{loading}$ is the catalyst loading per geometric surface area. For Fe/GD and Ni/GD, the $M_{loading}$ are 0.00221 and 0.00249 mg cm^{-2} , respectively.

(5) In our experiments, the saturated calomel electrode (SCE) was calibrated daily with respect to RHE. The calibration was carried out in the high purity hydrogen (H_2) saturated 0.5 M H_2SO_4 with a Pt foil as the working electrode. Cyclic voltammetry (CV) measurements were carried out at a scan rate of 1 mV s^{-1} . The average of the two potentials at which the current crossed zero was taken to be the thermodynamic potential for the hydrogen electrode reactions. This has been provided as supplementary material (Supplementary Fig. 15) in the revised version.

Supplementary Figure 15: Calibration of the saturated calomel electrode (SCE). In all measurements, SCE was calibrated with respect to RHE. The calibration was performed in the high purity hydrogen saturated electrolyte with a Pt foil as the working electrode. Cyclic voltammetry (CV) was run at a scan rate of 1 mV s^{-1} , and the average of the two potentials at which the current crossed zero was taken to be the thermodynamic potential for the hydrogen electrode reaction. **(a)** This SCE was used in the durability tests of Ni/GD in 0.5 M H_2SO_4 , $E(\text{RHE}) = E(\text{SCE}) + 0.257 \text{ V}$. **(b)** This SCE was used in other electrochemical tests in 0.5 M

H_2SO_4 , $E(\text{RHE}) = E(\text{SCE}) + 0.242 \text{ V}$.

4. On Page 14 line 244, the authors claimed that Fe/GD exhibits the smallest overpotential of only 9 mV. However, it is hard to see that in Figure 5. Even in Figure 5d, the onset potential is more likely at 50 mV. The authors should explain more about how this value of overpotential is obtained. Otherwise, the authors should correct this value and revise the phrase “close to 0 mV” in Page 4 line 76. The XANES characterization of Fe(Ni)/GD before and after HER measurements should also be performed in order to determine whether the atomic structure of Fe or Ni is changed during the HER measurement.

Response: The onset overpotential of catalysts is determined according to Tafel slopes. Corresponding data have been provided in the supplementary material (Supplementary Fig. 16). The as-prepared Fe/GD exhibits a small onset overpotential of ~ 9 mV. The XANES characterization of Fe(Ni)/GD before and after HER measurements were provided and added in revised manuscript as Supplementary Figure 17. The results indicated that the atomic structures of Fe/GD and Ni/GD showed no changed during the HER measurements.

Some discussions have been added in the revised manuscript as below.

Page 14 line 256-258:

There is no difference can be found in XANES and EXAFS spectra (Supplementary Figure 17) of Fe/GD and Ni/GD before and after HER measurements, indicating no change in the atomic structures occurred during the HER measurements.

Supplementary Figure 16: Tafel plot in the region of low current densities of (a) Fe/GD and (b) Ni/GD in 0.5 M H₂SO₄. The onset overpotential is determined by the potential when the plot starts to deviate from the linear region as indicated by the arrow.

Supplementary Figure 17: XAS studies before and after HER measurements. (a) *Ex situ* EXAFS and (b) the normalized XANES spectra at the Ni K edge of Ni/GD obtained before

(red line) and after (green dashed line) HER measurements. **(a)** *Ex situ* EXAFS and **(b)** the normalized XANES spectra at the Fe K edge of Fe/GD obtained before (red line) and after (green dashed line) HER measurements. Ni foil was measured for comparison.

Thank you again for your positive comments on our manuscript. Should there been any other corrections we could make, please feel free to contact us.

Yours Sincerely,

Yuliang Li

REVIEWERS' COMMENTS:

Reviewer #1 (Remarks to the Author):

The authors addressed the requested points and thus i recommend it for publication.

Reviewer #2 (Remarks to the Author):

Anchoring single-atom Ni(0) and Fe(0) on graphdiyne for hydrogen evolution

The questions and concerns in previous revision requests were properly addressed in the current version of manuscript. The authors have improved the quality of Raman and electrochemical measurements figures, and further added the required data including pre-edge derivative of XANES spectra. As graphdiyne is a less explored material in electrocatalysis area and single-atom catalyst, both its preparation and HER performance are interesting for a broad society of researchers. The structural explanation for single-atom on graphdiyne is acceptable in this current form. Therefore, I recommend the acceptance of this manuscript in Nature Communication.

The detailed point-by-point responses to each comment are listed as follows:

Reviewer #1 (Remarks to the Author):

The authors addressed the requested points and thus i recommend it for publication.

Response: Thank you very much for your assistants

Reviewer #2 (Remarks to the Author):

Anchoring single-atom Ni(0) and Fe(0) on graphdiyne for hydrogen evolution

The questions and concerns in previous revision requests were properly addressed in the current version of manuscript. The authors have improved the quality of Raman and electrochemical measurements figures, and further added the required data including pre-edge derivative of XANES spectra. As graphdiyne is a less explored material in electrocatalysis area and single-atom catalyst, both its preparation and HER performance are interesting for a broad society of researchers. The structural explanation for single-atom on graphdiyne is acceptable in this current form. Therefore, I recommend the acceptance of this manuscript in Nature Communication.

Response: Thank you very much for your assistants.

Thank you again for your positive comments on our manuscript. Should there been any other corrections we could make, please feel free to contact us.

Yours Sincerely,

Yuliang Li